# Comparative Proteomic Analysis of Protein Patterns of *Stenotrophomonas maltophilia* in Biofilm and Planktonic Lifestyles

**DOI:** 10.3390/microorganisms11020442

**Published:** 2023-02-09

**Authors:** Giovanni Di Bonaventura, Carla Picciani, Veronica Lupetti, Arianna Pompilio

**Affiliations:** 1Department of Medical, Oral, and Biotechnological Sciences, G. d’Annunzio University of Chieti-Pescara, Via dei Vestini, 31, 66100 Chieti, Italy; 2Center for Advanced Studies and Technology, G. d’Annunzio University of Chieti-Pescara, Via L. Polacchi 11, 66100 Chieti, Italy

**Keywords:** *Stenotrophomonas maltophilia*, biofilm formation, proteomic analysis, cystic fibrosis

## Abstract

*Stenotrophomonas maltophilia* is a clinically relevant bacterial pathogen, particularly in cystic fibrosis (CF) patients. Despite the well-known ability to form biofilms inherently resistant to antibiotics and host immunity, many aspects involved in *S. maltophilia* biofilm formation are yet to be elucidated. In the present study, a proteomic approach was used to elucidate the differential protein expression patterns observed during the planktonic-to-biofilm transition of *S. maltophilia* Sm126, a strong biofilm producer causing chronic infection in a CF patient, to identify determinants potentially associated with *S. maltophilia* biofilm formation. In all, 57 proteins were differentially (3-fold; *p* < 0.01) expressed in biofilm cells compared with planktonic counterparts: 38 were overexpressed, and 19 were down-expressed. It is worth noting that 34 proteins were exclusively found in biofilm, mainly associated with quorum sensing-mediated intercellular communication, augmented glycolysis, amino acid metabolism, biosynthesis of secondary metabolites, phosphate signaling, response to nutrient starvation, and general stress. Further work is warranted to evaluate if these proteins can be suitable targets for developing anti-biofilm strategies effective against *S. maltophilia.*

## 1. Introduction

*Stenotrophomonas maltophilia* is an environmental microorganism associated with plant rhizosphere and involved in the nitrogen and sulfur cycles [1]. However, as a human opportunist, it represents an emerging pathogen of significant concern to susceptible patient populations, especially the elderly and immunocompromised persons [2]. Indeed, this microorganism can cause many opportunistic infections, including those of the respiratory tract, bloodstream and heart, nervous system, gastrointestinal tract, urinary tract, bone, and soft tissue [3].

In addition to these nosocomial and community-acquired infections, the chronic colonization of the airways of patients with cystic fibrosis (CF) is also increasingly reported. With about a 10% colonization rate in these patients, *S. maltophilia* represents an independent risk factor for pulmonary exacerbation associated with impaired lung function, lung transplantation need, and death [4].

Even though it is considered a low-grade pathogen, the increased occurrence of *S. maltophilia* nosocomial infections is primarily the result of alarmingly high drug resistance rates, further complicated by biofilm formation, a relevant virulence feature [3,5]. *S. maltophilia* can adhere to several surfaces and grow as a biofilm, a microbial consortium embedded in a self-produced polymeric matrix and inherently resistant to antibiotic therapy and the host immune response [6]. Several studies have recently indicated that the ability to form biofilm is highly conserved in clinically relevant *S. maltophilia* isolates [7]. Biofilm formation can occur on inert surfaces (e.g., respiratory tubes and nebulizers, intravenous cannulae, prosthetic devices, and dental unit waterlines), causing medical implant-associated infections [3]. Regarding the biotic surfaces, *S. maltophilia* adheres to HEp-2 cells, mucin, and tight junctions of human epithelial respiratory cells [4]. Interestingly, it colonizes bronchial CF-derived cells as biofilms [8], highlighting the significance of biofilm formation in the persistence of *S. maltophilia* in the airways of CF patients because of a selective adaptation to CF airways.

For these reasons, *S. maltophilia* biofilm research has recently accelerated using genomics profiling to understand better the mechanisms underlying biofilm formation. The genetic regulation of biofilm formation in *S. maltophilia* is complex, as suggested by the significant number of genes involved [6]. Several studies indicated that the ability of *S. maltophilia* to form biofilm is mainly influenced by bacterial virulence factors (e.g., flagella, pili, LPS/exopolysaccharide biosynthesis, motility) [7], cell surface traits (e.g., outer membrane proteins, hydrophobicity) [9,10], and quorum-sensing communication [11].

However, as already observed for other bacterial pathogens, *S. maltophilia* biofilm cells show gene expression patterns significantly different from those of their planktonic counterparts [12]. The findings from bacterial transcriptomic analysis do not necessarily correlate with the detected proteins and their functionality, particularly in the case of CF strains in which the low genotype-phenotype correlation is probably the “biological cost” bacteria must pay to adapt to a highly stressful environment such as CF lung [13]. In this respect, the proteomics approach can provide a glimpse into the presence of functional molecules.

Despite the wealth of information derived from genetic studies on biofilm formation [6], studies have yet to compare the protein expression profiles between biofilm and planktonic *S. maltophilia* cells. To gain insights into the underlying mechanisms of *S. maltophilia* biofilm formation, in the present work, we used a proteomic approach to assess the changes in protein expression of a CF *S. maltophilia* strain when growing in biofilm versus planktonic lifestyles. The findings from the present study will help deliver comprehensive knowledge about the cellular processes and metabolic pathways involved in *S. maltophilia* biofilm to come up with potential targets useful to biofilm-inhibiting strategies. In this regard, the comparative analysis of the proteomes showed distinct differences between the protein profiles and related metabolic pathways.

## 2. Materials and Methods

### 2.1. Bacterial Strain and Growth Conditions

The *S. maltophilia* Sm126 strain, isolated from the airways of a CF patient with chronic infection, was used throughout this study. This strain was previously characterized as multidrug-resistant and with a propensity to produce a relevant biofilm [14].

Planktonic and biofilm cells for proteomic analysis were prepared from a standardized inoculum. Briefly, some colonies grew overnight on Muller-Hinton agar (Oxoid SpA; Garbagnate M.se, Milan, Italy) were suspended in 40 mL Trypticase Soy broth (TSB; Oxoid SpA) and incubated overnight at 37 °C, under agitation (130 rpm). Then, the broth culture was diluted with TSB to an OD_550_ of 1.5 and added to 560 mL TSB. The resulting 600 mL suspension was incubated at 37 °C under agitation (200 rpm) for 4 h to achieve a final OD_550_ of 0.6 (corresponding to 0.5–1 × 10^9^ CFU/mL).

### 2.2. Collection of Planktonic Cells

A volume of 150 mL of the standardized inoculum was incubated at 37 °C for 6 h, under agitation (200 rpm), when an exponential growth phase was achieved, corresponding to an OD_550_ of 0.6 (0.5−1 × 10^9^ CFU/mL). Planktonic cells were then harvested by centrifugation (10,000× *g*, 10 min, 4 °C), and the obtained pellet was washed thrice with Phosphate Buffer Solution (PBS) pH 7.3 (Merk Life Science Srl; Milan, Italy) before being subjected to lysis.

### 2.3. Collection of Biofilm Cells and Kinetics of Biofilm Formation

Sixty milliliters of the standardized inoculum were dispensed in each of five polystyrene, 150 mm diameter, tissue culture (TC)-treated Petri dishes (Iwaki). The plates were statically incubated at 37 °C for 24 h, then the biofilm samples were washed (3 times with PBS pH 7.3) and scraped to collect biofilm cells in a polypropylene vial (Nalgene; Thermo Scientific Italia) containing 150 mL of PBS. This suspension was centrifuged (10,000× *g*, 10 min, 4 °C), and the obtained pellet was washed three times with PBS and finally subjected to lysis.

The kinetics of biofilm formation in 96-well polystyrene microplates was spectrophotometrically monitored over 72 h using a crystal violet-based colorimetric assay [9].

### 2.4. Bacterial Lysis and Protein Extraction

Lysis was carried out using a Q Proteome Bacterial Protein Prep kit (Qiagen srl; Milan, Italy) containing 100 mg/mL lysozyme. Benzonase 0.25 U/µL (Sigma-Aldrich srl; Milan, Italy) was then added to degrade nucleic acids. Samples were incubated at room temperature on ice, then centrifuged (14,000× *g*, 30 min, 4 °C); the supernatant was collected in polypropylene vials (Nalgene) and newly centrifuged (50,000× *g*, 30 min). Aliquots of the supernatant underwent to SpeedVac (mod. SC110-SAVANT) and were finally stored at −80 °C until protein purification. Before purification, the three samples obtained by independent extractions were pooled for planktonic and biofilm samples [15]. Samples were purified using a Ready Prep 2D Cleanup kit (Bio-Rad; Milan, Italy), and the protein precipitates were treated with 50 µL solubilization buffer for two-dimensional gel electrophoresis (2-DE) (2 M thiourea, 7 M urea, 50 mM DTT, 4% CHAPS, 0.2% Bio-Lyte 3/10 ampholyte, and 0.002% bromophenol blue). After protein resuspension, each sample was centrifuged at 12,000× *g*, and the clear phase was transferred to a new vial where the buffer was added to yield a final volume of 450 µL. Protein concentration was determined, according to Lowry, by RC-DC Protein Assay (Bio-Rad). Concentrations of 8 and 16 µg/µL were obtained for planktonic and biofilm samples, respectively. Samples were finally stored at −20 °C until needed.

### 2.5. 2-D Electrophoresis

Immobilized pH gradient precast Ready Strips with a non-linear gradient (pH 3 to 10) (Bio-Rad) were rehydrated for 12 h with 80 Wg protein in 8 M urea, 2 mM tributyl phosphine, 2% ampholytes pH 3–10, 2% CHAPS and traces of bromophenol blue. The first dimension was carried out using the Protean IEF Cell (Bio-Rad) for 63.7 kVh. After isoelectric focusing, the strips were first equilibrated for 15 min in equilibration buffer (6 M urea, 0.375 M Tris pH 8.8, 2% SDS, and 20% glycerol). The same solution with 2.5% iodoacetamide was used for further 15 min equilibration period. The second-dimensional separation was a vertical SDS-PAGE with 12.5% acrylamide resolving gels in Protean II XL (Bio-Rad). Separation was performed at 20°C under constant amperage (40 mA). Gels were stained with 0.15% colloidal Coomassie blue G-250 (Sigma-Aldrich srl; Milan, Italy) for 48 h, then scanned using a GS-800 Imaging densitometer (Bio-Rad).

### 2.6. Analysis of Protein Patterns

Six gels resulting from three independent protein extractions were comparatively analyzed for each condition (planktonic, biofilm). Only reproducible spots present in four to six gels were considered in the analysis. Gels were analyzed with PD-Quest software (Bio-Rad; ver. 7.0) for qualitative and quantitative analysis of protein spots. Statistical analysis was performed by GraphPad (GraphPad Software Inc.; Boston, MA, USA): Student’s *t*-test was carried out to ensure significant (*p* < 0.05) changes in the value of protein spots. Calibration of gels with an isoelectric point (pI) and molecular masses was carried out using internal 2-D SDS-PAGE protein standards (Bio-Rad).

### 2.7. Protein Identification by MALDI-TOF-MS and N-Terminal Sequencing

The 2-D gel protein spots, stained with Coomassie blue, were digested with 0.5 µg/µL trypsin. Peptides were mixed with a 10 mg l31 solution of K-cyano-4-hydroxycinnamic acid, placed on the sample plate to air-dry, then analyzed by MALDI-TOF-MS (Reflex IV, Bruker Daltonics, Brehme, Germany). Monoisotopic peptide masses obtained from mass spectra were acquired, and proteins were identified by matching the sequences derived from peptide MS/MS spectra with sequences in the *S. maltophilia* protein sequence database (https://www.ncbi.nlm.nih.gov/protein; accessed on 1 February 2015) using Mascot Daemon database searching software (MatrixScience; ver. 2.7). The following parameters were used in the searches: protein molecular mass range from 1000 to 100,000 Da, trypsin digest with one missing cleavage, fragment ion mass tolerance of T 50 ppm, and possible methionine oxidation. N-terminal amino acid sequences were obtained by transferring the selected proteins to polyvinylidene difluoride membranes (Sequi-Blot PVDF Membrane, Bio-Rad), and then micro-sequenced using an automatic Beckman/Porton LF3000 protein sequencer. Sequence homology was evaluated with the BLAST program for MALDI-TOF-MS and N-terminal sequencing identification.

### 2.8. Scanning Electron Microscopy

Biofilm samples were allowed to form onto 35-mm TC-treated polystyrene dishes (Iwaki), air-dried overnight, and then fixed for 16 h in 2% (*v*/*v*) paraformaldehyde and 2% (*v*/*v*) glutaraldehyde in 0.15 M sodium cacodylate buffer pH 7.4, added with the cationic dye 0.1% alcian blue (Polysciences Europe, Germany). Samples were rinsed three times in 0.2 M cacodylate buffer for 10 min and post-fixed for 1 h in 1% OsO4 (*v*/*v*). Samples were rinsed in 0.15 M cacodylate buffer and then dehydrated in a graded ethanol series (50, 70, 80, 95, and 100%) before critical point drying. Samples were mounted on aluminum stubs, coated with 15-nm Au film, and then observed with a Philips XL30CP SEM.

## 3. Results and Discussion

### 3.1. Growth Kinetics of Planktonic and Biofilm S. maltophilia Cells

To standardize the preparation of samples for proteomics analysis, growth curves of planktonic and biofilm cells were obtained under the experimental conditions used, as shown in Figure 1. The growth of planktonic cells was spectrophotometrically monitored over 48 h, as previously described [13]. The growth curve analysis revealed an exponential phase entry after 4 h-incubation (OD_550_: 0.46), lasting up to 8 h (OD_550_: 2.02) (Figure 1A). Therefore, planktonic cells for proteomic analysis were collected following 6 h-incubation when an OD_550_ value of ~1.33 (corresponding to ~1 × 10^9^ CFU/mL) is reached as indicative of the mid-exponential phase.

The kinetics of biofilm formation on polystyrene is shown in Figure 1B. The biofilm biomass increased over time until 24 h-incubation when a plateau was reached (OD_550_: 0.263), then gradually decreasing towards 72 h. As a result, the biofilm cells for proteomic analysis were collected after 24 h-incubation when the biofilm differentiated into a “mature” structure consisting of a multilayer architecture embedded in large amounts of extracellular matrix appearing as fine polymeric fibrils bridging the cells, as shown by scanning electron microscopy analysis (Figure 2).

### 3.2. Proteomic Analysis: Biofilm versus Planktonic Lifestyles

To identify the differentially expressed proteins in *S. maltophilia* during the transition from planktonic to sessile lifestyles, the proteomic profiles expressed under these conditions were compared using 2-DE, and peptide sequences were identified by MALDI-TOF. The identification of proteins of interest was then carried out by research in the sequenced genome of *S. maltophilia* and by analysis of sequence homology with known proteins of other microorganisms. Although the genome of *S. maltophilia* has been sequenced, the functions of most genes have yet to be defined [16].

The image analysis showed approximately 350 spots/gels after staining with colloidal blue Coomassie G-250, irrespective of the sample (planktonic or biofilm cells) considered (Figure 3). Most proteins migrated with pI between pH 4 and 8. High-reproducible gels with a correlation coefficient of at least 80% could be achieved. Among these 350 spots, the comparison of 2-DE proteomic maps in the considered pH range (3–10 non-linear) revealed 198 protein spots not differentially expressed between biofilm and planktonic cells. On the contrary, 149 spots displayed statistically significant differences (at least 1.5-fold; *p* < 0.05, *t*-test) in their level of protein expression between biofilm and planktonic growth. Particularly, 76 in biofilm cells (42 upregulated and 34 exclusive proteins) (Appendix A), and 73 in planktonic cells (56 upregulated and 17 exclusive proteins) (Appendix A).

To strengthen the comparison between the biofilm and planktonic growth states among the 149 differently expressed spots, only those displaying a signal >1000 ppm, and with a difference of ≥ 3-fold between biofilm and planktonic growth (*p* < 0.01, *t*-test) in the signal intensity were selected and underwent identification by MALDI-TOF-MS and N-terminal sequencing. Specifically, 34, 19, and 38 spots were respectively exclusive, upregulated, or downregulated in biofilm cells. Not all the selected spots could be identified due to technical reasons (i.e., no sequence found due to the incomplete genomic database or the lacking sequence for the strain under study; lack of a sufficient amount of available protein). In addition, some proteins occurred as isoforms, and most were characterized by small mass changes but clearly different pI values.

### 3.3. Proteins and Related Pathways Expressed Only in Biofilm Cells

Sixteen proteins, among those present only in biofilm cells, were identified by sequence homology with those deposited in databases (software Mascot, ver. 2.7; Matrix Science): fourteen in the *S. maltophilia* database, while two were matched with those in the database of other bacteria (i.e., B1305 from *Streptomyces sviceus*, B3102 from *Geobacter* spp.) (Table 1).

#### 3.3.1. Quorum Sensing-Mediated Intercellular Communication

Gram-negative bacteria modulate and synchronize gene expression involved in pathogenicity through a mechanism based on the population density called “quorum sensing” (QS) [17]. It acts by the production and perception by the bacterial population of one or more diffusible “signal” molecules whose concentration reflects cell density. The signal reaching a threshold concentration is perceived by a sensor system that activates the QS [17]. The most frequently produced signals from Proteobacteria have been grouped in the class of N-acyl homoserine lactones (N-AHSLs), almost always synthesized through a LuxI-like synthase and perceived through a sensor-regulator system LuxR-like [18]. In *S. maltophilia*, the main QS system relies on the Diffusible Signal Factor (DSF) cis-11-methyl-2-dodecenoic acid, which modulates motility, biofilm formation, antibiotic resistance, and virulence [19]. DSF produced by *S. maltophilia* influences several virulence traits of *P. aeruginosa*—e.g., biofilm formation and antibiotic resistance—and its persistence in CF lungs [20]. On the other hand, although *S. maltophilia* cannot produce AHLs, it also responds to AHL signal molecules produced by *P. aeruginosa* [21].

The existence of QS makes it plausible that both the host and the bacteria could have developed “quorum quenching” (QQ) strategies to interfere with this intercellular communication system. Confirming this, N-AHSLs can be rapidly degraded in nature [22] by environmental pH and temperature or by bacteria, plants, and animals through the production of antagonists [23] or degrading enzymes such as lactonases [24] and starch hydrolases [25]. In this frame, we demonstrated for the first time that *S. maltophilia* specifically produces starch hydrolase (B5613 protein; *S. maltophilia* K279a) in the biofilm phenotype, suggesting the presence of a QQ system. Indeed, the cleavage of N-AHSLs by this enzyme might generate homoserine and an acyl chain that cannot spontaneously regenerate a functional QS signal.

This ability might have relevant implications in CF interspecies communication. *S. maltophilia* might indeed attenuate *P. aeruginosa* pathogenicity by a QQ-based mechanism, degrading N-AHL signal molecules that modulate several virulence traits of *P. aeruginosa*—e.g., biofilm formation and antibiotic resistance—and its persistence in CF lungs [20,26]. Further studies aimed at understanding the behavior of bacterial populations during CF infection and testing the therapeutic QQ antivirulence potential of *S. maltophilia* biofilms are warranted.

#### 3.3.2. General Stress Response and Nutrient Starvation

Our findings showed that some proteins exclusively expressed in *S. maltophilia* Sm126 biofilm-grown cells are involved in nutrient limitation and general stress response, including a putative heat shock protein (B6109 protein), the acetyl-CoA synthetase (B3712 protein), and chorismate-synthetase (B3102 protein).

##### Putative Heat Shock Protein (B6109 Protein)

Universal stress proteins are expressed in nearly all bacteria, playing a role in adaptation to oxidative stress, low pH, high temperature, and hypoxia [27]. Heat shock proteins (HSPs) are implicated in the bacterial response to environmental stresses and the pathogenesis of infection. For example, in *P. aeruginosa* the HSP DNA has a significant effect on pathogenicity, promoting bacterial adhesion and biofilm formation [28]. This effect is mainly due to a decreased extracellular DNA—one of the most critical components in biofilms—secondary to the DNA-mediated reduction of the QS-autoinducer 2-heptyl-3-hydroxy-4(1H) -quinolone [28].

##### Acetyl-CoA Synthetase (B3712 Protein)

To survive, many cells must be able to modulate their physiology, growing rapidly in the presence of abundant nutrients and then slowdown in nutrient deficiency. During this transition, the bacteria pass from a program of rapid growth, which produces and frees the acetate (dissimilation) in the environment, to a program of slow growth facilitated by the uptake and utilization (assimilation) of the acetate previously excreted. This phenomenon, called the “acetate switch”, occurs when cellular metabolism reduces the environmental content in acetate-producing sources (acetogenins), such as D-glucose or L-serine, and they begin to depend on their ability to recover environmental acetate [29]. The key molecule in the “acetate switch” is acetyl-coenzyme A (acetyl-CoA), a high-energy intermediate of the central metabolism. Recently there has been a renewed interest in this phenomenon resulting from the theory that acetyl-phosphate, a high-energy intermediate of the dissimilation pathway, can work similarly to a global signal [29]. In particular, considerable evidence supports the regulatory role of acetyl-phosphate in numerous cellular processes. Acetyl-phosphate is associated with a reduction in gene expression involved in flagellar synthesis [30]. On the other hand, it causes increased gene expression related to the assembly of pilus type I, the synthesis of colonic acid (extracellular polysaccharide or capsule), and the response to multiple stresses. It has also been suggested that acetyl-phosphate coordinates the expression of surface structures and cellular processes involved in biofilm formation [31]. It is fascinating, in this regard, the recent observation that the formation and disintegration of structures such as biofilm, formed intracellularly by *Escherichia coli* uropathogens, depend on the same cellular structures regulated by acetyl-phosphate, such as flagella, pili-type 1 and capsule [32]. The induction of acetyl-CoA synthetase in the biofilm of *S. maltophilia* could, therefore, be the consequence of the “acetate switch” during the cellular transition from a fast-growing physiological program (planktonic forms) to one characterized by cellular quiescence (sessile forms formed biofilm).

##### Chorismate-Synthetase (B3102 Protein) and Polyribonucleotide phosphorylase (B3808 Protein)

It has been suggested that under low levels of folic acid, some enzymes of the shikimate pathway, aimed at the biosynthesis of folates and aromatic amino acids, may be considered excellent targets for antibiotic therapy since, essential in bacteria, fungi, and other plants, such enzymes are not present in animal cells. The chorismate-synthetase is the seventh enzyme of this pathway where it catalyzes the NADH- and FMN-dependent synthesis of chorismate, a precursor of aromatic amino acids (phenylalanine, tyrosine, and tryptophan), naphthoquinones (plant pigments with antibiotic and immunomodulatory activity), and menaquinones (vitamins K). It has recently been seen that cyanobacteria are provided with QS and release the mediator C8-AHL (N-octanoyl homoserine lactone). In response to C8-AHL, during the self-induction phenomenon, cyanobacteria have been found to increase their levels of chorizo-synthase [33] significantly. The chorismate-synthase observed exclusively in the biofilm phenotype of *S. maltophilia* again supports the hypothesis that this microorganism is also equipped with a QS N-AHSL.

Polyribonucleotide phosphorylase (PNPase) is a conserved enzyme in bacteria, plants, and mammals where it affects gene expression [34]. It is involved in two reactions [35]: (i) the processive phosphorolytic degradation of mRNA from the 3′-end, and (ii) a polymerase activity catalyzing reaction using nucleoside diphosphates as substrates to add NMPs to the 3′-end of the RNA chain. In addition, PNPase can be directly or indirectly involved in several functions, such as virulence, stress responses, biofilm formation, and motility [35]. In *Salmonella enterica* Serovar Typhimurium, PNPase negatively controls biofilm formation by affecting CsgD expression [36]. At the same time, in *E. coli* it represses poly-N-acetylglucosamine production [37] and the expression of protein 43, an outer membrane protein promoting cell aggregation and affecting motility [38]. Considering that PNPase negatively modulates biofilm formation, the exclusive production we observed in *S. maltophilia* biofilms warrants further investigation.

#### 3.3.3. Carbohydrate Metabolism and Energy Production

Other proteins exclusive to *S. maltophilia* biofilm are involved in sugar metabolism and energy generation

The induced expression of 2,3-bisphosphoglycerate-dependent phosphoglycerate mutase and phosphoglycerate mutase (B6314 and B7203 proteins) in biofilm samples confirmed earlier findings indicating that this glycolytic enzyme (GpmA) modulates biofilm formation in *S. maltophilia* on airway epithelial cells and abiotic surfaces [39]. It has been hypothesized that amino acid carbons are shuttled into the central metabolism through GpmA and toward pentose phosphate metabolism, producing an unknown compound able to stimulate growth and biofilm [40]. A greater understanding of these unique metabolic pathways is needed to define novel methods for inhibiting *S. maltophilia* growth and biofilm. This enzyme is also involved in the fermentative pathway. It is well known that oxygen tension is significantly reduced in the deeper layers of the biofilm [41] and that *E. coli* gene expression is altered due to reduced oxygen availability [42]. For this reason, the expression of phosphoglycerate-mutase could be a consequence of the poor oxygenation in the biofilm formed by *S. maltophilia*. On the other hand, glycolytic enzymes have been shown to have additional properties when located at the cell surface. For example, streptococcal α-enolase can bind plasminogen highly when associated with cell surface [43]. Similarly, in *Staphylococcus aureus,* a superficial protein, the glycolytic glyceraldehyde-3-phosphate dehydrogenase enzyme, with high transferrin affinity, has been identified. Therefore, the glycolytic pathway is one of the essential factors among biofilm survival mechanisms in *S. maltophilia*. As in other microorganisms, the expression of glycolytic enzymes in *S. maltophilia* biofilm can be regulated, not only by oxygen but also by other stimuli, thus indicating they play a role in a complex microbial population.

The second protein exclusively expressed in biofilm cells was identified as isocitrate dehydrogenase (B4409 protein) (IDH), a key enzyme in the tricarboxylic acid cycle, which is involved in biofilm formation by *Bacillus cereus* and *S. aureus*. In both species, the complete loss of IDH reduces biofilm yield and changes its morphology, while the growth seems not to be affected [44,45]. Specifically, in *S. aureus,* the phosphorylation of IDH resulted in the complete loss of activity and was restored upon dephosphorylation [45]. It has been suggested that IDH, in *B. cereus* and *S. aureus*, may regulate biofilm formation acting on intracellular redox homeostasis [44,45]. No studies have been published in this regard on *S. maltophilia*. However, the induced expression of the hypothetical oxidoreductase protein SACOL1851 we observed might indicate a similar mechanism also in *S. maltophilia*.

#### 3.3.4. Other Pathways

Regarding the hypothetical protein Smal (B7007 protein), although the structure of this protein has not been determined previously, it is probably a phosphate-binding protein from *S. maltophilia* (UniProt ID B4SL31; gene ID Smal_2208). Because biofilm formation is commonly controlled by phosphate signaling, SmeI might play a role in biofilm formation by mediating the cellular uptake of phosphate ions [46]. In *P. aeruginosa*, a functionally similar protein, PstS, has been shown to contribute to the structural integrity of biofilm, even though phosphate binding per se is not required for PstS biofilm activity [47]. Future studies are needed to characterize how SmaI contributes to biofilm formation in *S. maltophilia*, and whether it may represent a novel target for antibiofilm strategies.

No studies were published on the involvement of other biofilm-related proteins—putative electron transfer flavoprotein-ubiquinone oxidoreductase (B6708), putative 3-ketoacyl-CoA thiolase (B8509), conserved hypothetical exported protein (B8812), putative 4-aminobutyrate-2-oxoglutarate aminotransferase (B5617), 2,3,4,5-tetrahydropyridine-2,6-dicarboxylate N-succinyltransferase (B8206), and putative dihydrolipoamide succinyltransferase E2 component (B9202)—in bacterial biofilm formation. Therefore, the function of these proteins in *S. maltophilia* biofilm production remains unclear, thus warranting further studies.

### 3.4. Proteins Significantly Upregulated in Biofilm Cells

Nine protein spots upregulated in the biofilm cells were identified, all by searching the *S. maltophilia* database (Table 2).

#### 3.4.1. Putative Urocanate Hydratase

The first three enzymatic steps by which organisms degrade L-histidine to L-glutamate eventually lead to purine/pyrimidine biosynthesis via carbamoyl-phosphate and carbamoyl-aspartate, which are universally conserved in many species from all kingdoms of life. A histidine ammonia-lyase catalyzes 1,2-elimination of the α-amino group from l-histidine; a urocanate hydratase converts urocanate to 4-imidazolone-5-propionate, and this intermediate is hydrolyzed to N-formimino-l-glutamate by an imidazolone-propionase [48]. In *S. aureus*, the endogenous production of glutamate—a major amino acid in central metabolism—enhances biofilm formation in conjunction with the TCA and urea cycles [49]. In other work, Hassanov et al. [50] showed that the development of *Bacillus subtilis*, *Enterococcus faecalis,* and *P. aeruginosa* biofilms depends specifically on the use of glutamine or glutamate as a nitrogen source, contrarily to planktonic cells. Specifically, in *B. subtilis* biofilms, glutamine is necessary only for the dividing cells at the edges, while the inner cells use lactic acid, thus indicating a differential metabolic requirement within the biofilm. Our results confirmed that glutamine is an essential trigger for biofilm development also in *S. maltophilia*; targeting the metabolism might lead to novel anti-biofilm therapies.

#### 3.4.2. NADP-Dependent Alcohol Dehydrogenase

ADH is one of the dehydrogenase enzymes occurring in many organisms to catalyze ethanol metabolism. An increasing number of evidence demonstrated that ADH is also involved in the bacterial QS system playing a crucial role in bacteria biofilm formation. Both in clinical and reference, *Acinetobacter baumannii*, *S. aureus*, and *P. aeruginosa* strains ADH leads to dose-dependent increases in bacterial growth, motility, and biofilm development, along with an increased expression of QS-related genes [51]. In another work, Becker et al. [52] reported upregulated in *S. aureus* biofilms by ADH1. Contrarily, the disruption of ADH1 significantly enhanced the ability of *Candida albicans* to form biofilm. The differential role of this enzyme in different microbial biofilms may have implications for the interactions in a mixed-species milieu. It is likely that bacteria overexpress ADH not only to enhance their ability to form biofilm, but also to inhibit *Candida* biofilm formation. These findings suggest that ADH may represent a new target to control biofilm-associated *S. maltophilia* infections.

#### 3.4.3. Serine Hydroxymethyltransferase

Serine hydroxymethyltransferase (SHMT) is a pyridoxal 5′-phosphate (PLP)-dependent enzyme. In *P. aeruginosa*, the reversible conversion of the amino acids glycine to serine by hydroxymethyltransferase (SHMT) impacts the pathogen’s sessile lifestyle and virulence. The inactivation of SMHT leads to the rugose small-colony variants phenotype of *P. aeruginosa* and increases biofilm formation and exopolysaccharide synthesis while decreasing swarming motility [53]. SHMT controls biofilm formation by regulating the second messenger cyclic diguanylate (c-di-GMP). Whole transcriptome analysis also revealed that SHMT is associated with the cell’s redox state [53]. The upregulation of SHMT expression we found in biofilm cells suggests a differential role of the enzyme in *S. maltophilia*, although we do not know the underlying mechanisms.

#### 3.4.4. Acetyl-CoA Carboxylase Biotin Carboxylase Subunit

Acetyl-CoA carboxylase (ACC) plays a critical role in fatty acid metabolism, catalyzing the first committed and rate-limiting step in fatty acid biosynthesis. In *E. coli*, the ACC reaction consists of two partial reactions catalyzed by different components of a multi-subunit complex. In the first reaction, the *accC*-encoded biotin carboxylase subunit allows the carboxylation of biotin by bicarbonate in an ATP-dependent reaction to form carboxybiotin [54]. The increased AccC expression we observed in biofilm cells might be QS-related. Indeed, in *S. maltophilia* the QS system is based on the fatty acid signal DSF (*cis*-11-methyl-2-dodecenoic acid) [55] that positively regulates biofilm formation and the induction of L1 and L2 β-lactamase production [56]. In addition, the presence of DSF signals in CF sputum was associated with the patient’s colonization exclusively by *S. maltophilia* and *B. cenocepacia*, thus, indicating the potential for interspecies signaling involving DSF family signals within the CF lung [57]. These findings indicate that acetyl-CoA carboxylase might represent a valuable antibiotic and antivirulence therapeutic target.

#### 3.4.5. Aminotransferase

Weiland-Bräuer et al. [58] recently identified nine QQ proteins that significantly inhibit biofilm formation of *E. coli* K12 MG1655, *P. aeruginosa, B. subtilis*, and *S. aureus* likely due to the interference with AI-2 and AHL-based QS. The sequence-based prediction analysis revealed aminotransferase among these novel QQ proteins.

No published studies reported the role of threonine 3-dehydrogenase, involved in the catabolism of threonine and the synthesis of glycine in *E. coli* [59], in biofilm formation. Finally, we observed a conserved hypothetical protein (NMB1475) upregulated in biofilm cells.

### 3.5. Proteins Significantly Downregulated in Biofilm Cells

Eight proteins, among those down expressed in biofilm cells, were identified by sequence homology with those deposited in databases (software Mascot, ver. 2.7; Matrix Science): 7 in the *S. maltophilia* database, and one (MP8009) in the database of *Campylobacter rectus* RM3267 (Table 3).

#### 3.5.1. OmpA/MotB Proteins

The proteins of the OmpA/MotB domains act as porin-like integral membrane proteins in most Gram-negative bacteria, which are required for cell morphology, membrane stability, and motility [60]. OmpA is the most abundant outer membrane protein in *S. maltophilia,* providing the structural integrity needed for flagellar assembly and swimming motility [61]. In *P. aeruginosa*, the major flagellar motor proteins MotA and MotB form a complex that acts as the stator and generates the torque that drives flagellar rotation. Therefore, the decrease in the expression of proteins of the OmpA/MotB domain observed in biofilm cells is consistent with our previous findings confirming that in *S. maltophilia* flagella are critical only during initial stages of attachment, while being turned off in mature biofilms [8,62].

#### 3.5.2. Thioester Dehydrase

Transcriptomic studies showed that in acute infection, planktonic *P. aeruginosa* cells rely on amino acids as a carbon source, while in biofilm infection sessile cells may utilize fatty acids for carbon and energy [63]. A similar pathway might be hypothesized in *S. maltophilia* based on the down expression of the thioester dehydrase family protein involved in the biosynthesis of fatty acids. In addition, the degradation of the fatty acids would improve acetyl-CoA production, as also suggested by the unique expression of acetyl-CoA synthetase we observed in sessile cells, potentially feeding the expression of surface structures, such as flagella, pili, and exo-polysaccharides, involved in the initial stages of biofilm development [43].

#### 3.5.3. Thioredoxin

The redox protein thioredoxin has been shown to modulate various virulence factors in Gram-negative pathogens. Previous research has shown that it is critical for the regulation of motility and gene transcription in *Listeria monocytogenes* [64], reduction of mucin to improve *Helicobacter pylori* adhesion to the epithelial surface [65], and contributes to organ burdens, host response, and increased survival in a model of sepsis caused by *A. baumannii* [66]. The decreased expression of thioredoxin we observed during the planktonic-to-biofilm transition might, therefore, indicate that bacteria lower their virulence by forming biofilm so that it can achieve persistent infection in vivo. Consistent with this hypothesis, previous studies reported the downregulation of genes encoding for virulence factors in biofilm cells relative to planktonic cells of several bacterial species [67,68,69]. Specific to CF, *P. aeruginosa* adaptation in CF airways selects patho-adaptive variants with a lower ability to cause acute infection [70]. Similarly, we previously showed that the virulence potential of *S. maltophilia* plays little if any role in its ability to persist in CF airways [71]; confirming this, the *S. maltophilia* Sm126 strain used in the present study was responsible for chronic infection in a CF patient.

#### 3.5.4. Cold Shock Protein

The cold shock protein (Csp) superfamily consists of homologous proteins highly conserved in many bacteria [72]. Although most research has focused on their role in bacterial adaption and survival at suboptimal temperatures, recent studies indicate that some Csps’ also contribute to general stress response and virulence. Specifically, numerous pieces of evidence indicated that Csp homologs positively regulate the motility and attachment to surfaces, as observed in *Xylella fastidiosa* [73], *L. monocytogenes* [74], and *Salmonella Typhimurium* [75]. These findings might explain the downregulation of Csp we observed in cells collected from a 24-h-old biofilm. Indeed, both motility and adhesion are needed in the early phase of biofilm formation, although being quenched in mature biofilm.

## 4. Conclusions

This work reports the first proteomic analysis of an *S. maltophilia* biofilm, highlighting that biofilm development results in significant changes in the whole-cell protein profiles of *S. maltophilia* growing planktonically and as a biofilm on polystyrene.

The proteins identified are implicated at various levels of cellular physiology, indicating that the biofilm phenotype results in complex patterns of gene regulation. In particular, the proteins we observed being exclusively associated with the biofilm phenotype—i.e., starch hydrolases (QS-mediated intercellular communication); a putative heat shock protein, acetyl-CoA synthetase, and chorismate-synthetase (response to stress and nutrient starvation); polyribonucleotide phosphorylase (phosphate signaling); 2,3-bisphosphoglycerate-dependent phosphoglycerate mutase, phosphoglycerate mutase, and isocitrate dehydrogenase (exalted carbohydrate metabolism)—may play an important role in the regulation of the biofilm development in S. maltophilia and, therefore, representing interesting targets to develop suitable anti-biofilm agents. Further functional and gene mutation studies are warranted to prove the exact role of the identified proteins in *S. maltophilia* biofilm development.

## Figures and Tables

**Figure 1 microorganisms-11-00442-f001:**
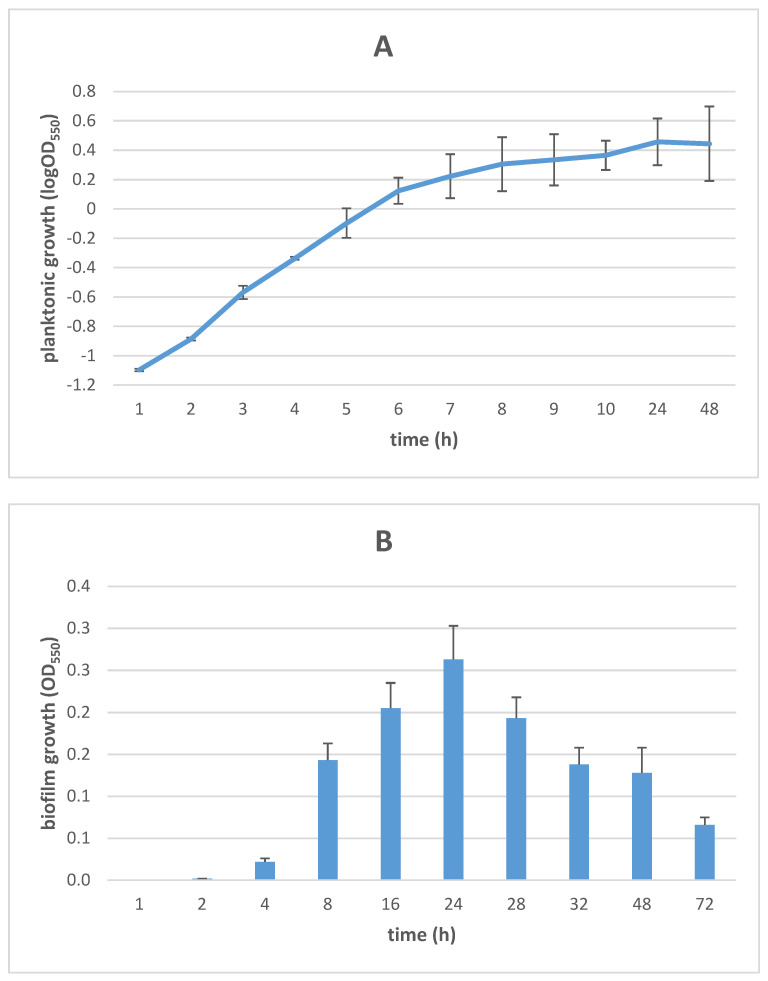
Growth kinetics of planktonic and biofilm *S. maltophilia* Sm126 cells. (**A**) Planktonic kinetics was measured in Trypticase Soy broth, under dynamic conditions, and results are shown as the log of mean OD_550_ ± SD values (n = 6). (**B**) Biofilm formation kinetics was measured in a 96-well polystyrene microplate using a crystal violet-based colorimetric assay, and results are shown as mean OD_550_ + SD values (n = 6).

**Figure 2 microorganisms-11-00442-f002:**
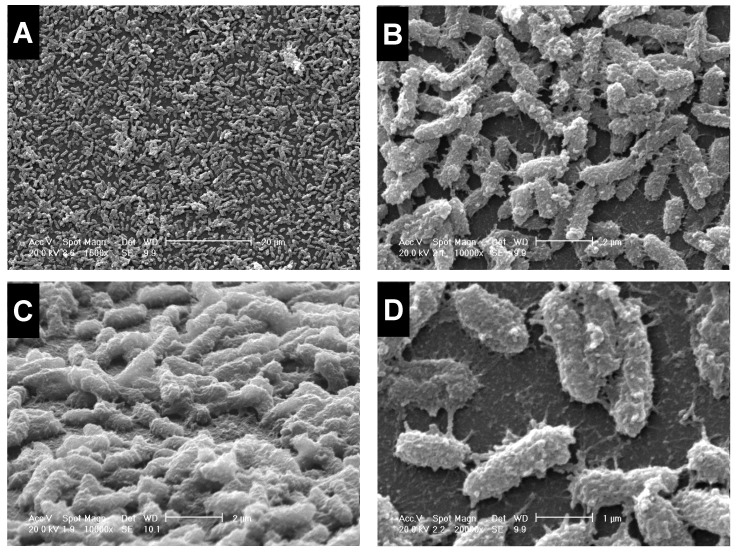
Scanning electron microscopy observation of *S. maltophilia* mature biofilm. The Sm126 strain was allowed to form biofilm onto 35-mm TC-treated polystyrene dishes for 24 h at 37 °C under static conditions. Magnifications: ×1.500 (**A**); ×10.000 (**B**); ×10.000 (**C**) (60°-inclined); ×20.000 (**D**).

**Figure 3 microorganisms-11-00442-f003:**
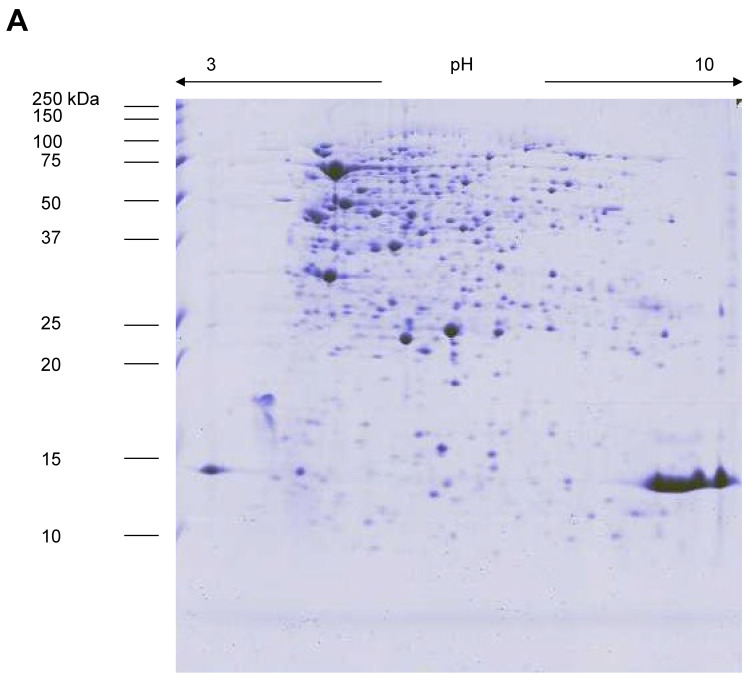
The 2-DE gel electrophoresis of *S. maltophilia* Sm126 proteins from biofilm (**A**) and planktonic (**B**) cells.

**Table 1 microorganisms-11-00442-t001:** Proteins (n = 16; signal >1000 ppm; *p* < 0.01, *t*-test) expressed only in biofilm-grown *S. maltophilia* Sm126 cells.

Spot No.	Protein	Biological Process	Molecular Function	Peptides Matched ^a^	Sequence Coverage ^b^	pI ^c^	MW ^d^	GI ^e^
B1305	regulatory protein	Not known	Not known	6	33	4.83/5.96	36,096/25,543	197784663
B5613	putative amidohydrolase	Nitrogen compound metabolic process	Hydrolase activity	11	31	5.65/6.38	49,616/52,720	190575191
B6708	putative electron transfer flavoprotein-ubiquinone oxidoreductase	Energy production and conversion	Electron-transferring-flavoprotein dehydrogenase activity; Metal ion binding	11	29	5.89/6.60	61,233/72,075	190572277
B8509	putative 3-ketoacyl-CoA thiolase	Fatty acid beta-oxidation	Acetyl-CoA C-acyltransferase	14	43	7.15/8.66	46,154/45,239	190572243
B7007	hypothetical protein Smal	Not known	Not known	8	48	6.09/7.19	16,432/11,577	194367327
B4409	putative isocitrate/isopropylmalate dehydrogenase	Leucine biosynthesis	Oxidoreductase activity; Mg and NAD binding	16	57	5.61/6.17	35,841/39,353	190573015
B3102	chorismate synthase	Aromatic amino acid family biosynthetic process	Lyase chorismate synthase activity	13	45	5.41/5.56	42,701/20,525	191161246
B3712	putative acetyl-coenzyme A synthetase	Acetyl-CoA biosynthesis from acetate	Metal ion binding; AMP/ATP binding; Acetate-CoA ligase activity	17	28	5.55/5.61	72,107/67,921	190576408
B6109	putative heat shock protein	Not known	Not known	10	74	5.74/6.61	19,550/17,758	190575907
B8812	conserved hypothetical exported protein	Not known	Catalytic activity	28	43	8.76/8.73	77,975/83,683	190576412
B3808	putative polyribonucleotide phosphorylase	RNA processing; mRNA catabolic processing; 3′-5-exoribonuclease activity; RNA binding; Polyribonucleotide nucleotidyl-transferase	Nucleotidyl-transferase	20	34	5.42/5.63	75,448/79,698	190575258
B7203	phosphoglycerate mutase 1 family	Gluconeogenesis; Glycolysis	Isomerase	9	54	6.51/7.24	28,036/26,920	194364976
B5617	putative 4-aminobutyrate-2-oxoglutarate aminotransferase	Alanine, glutamate, and glutamine metabolism	Pyridoxal phosphate binding; 4-amino-butyrate:2-oxoglutarate transaminase activity	9	26	5.82/6.58	52,261/54,309	190573417
B8206	2,3,4,5-tetrahydropyridine-2,6-dicarboxylate N-succinyltransferase	Lysine and diamonipimelate biosynthesis	2,3,4,5-tetrahydropyridine-2,6-dicarboxylate N-succinyltransferase activity	8	18	9.71/8.76	37,654/29,196	190573512
B9202	putative dihydrolipoamide succinyltransferase, E2 component	Tricarboxylic acid cycle	dihydrolipoyllysine-residue succinyltransferase activity	9	18	5.71/9.39	41,992/26,817	190575085
B6314	2,3-bisphosphoglycerate-dependent phosphoglycerate mutase	Gluconeogenesis; Glycolysis	2,3-bisphosphoglycerate-dependent phosphoglycerate mutase activity	11	61	6.32/6.93	28,062/27,123	190573432

^a^ Number of tryptic peptides matched to protein sequence based on MS/MS spectra. ^b^ Percentage of amino acid coverage (peptides observed/theoretical number from sequence data given in *S. maltophilia* database). ^c^ Theoretical and practical pI of the protein. ^d^ Theoretical and practical mass, expressed in Da, of the protein. ^e^ GI accession (GenInfo identifier; sequence identification).

**Table 2 microorganisms-11-00442-t002:** Proteins (n = 9; signal > 1000 ppm; ≥ 3-fold *vs*. planktonic growth; *p* < 0.01, *t*-test) significantly upregulated in biofilm-grown *S. maltophilia* Sm126 cells.

Spot No.	Protein	Biological Process	Molecular Function	Fold Up-Regulation ^a^	Peptides Matched ^b^	Sequence Coverage ^c^	pI ^d^	MW ^e^	GI ^f^
MB4713	Putative urocanate hydratase	Aminoacid degradation	Conversion of urocanate to 4-imidazolone-5-propionate	7.8	13	30	5.58/6.06	60,368/69,065	190575003
MB4807	Putative zinc metallopeptidase	Cell division; Protein catabolysis; Proteolysis	Metallopeptidase activity; Zinc ion binding	10.9	15	25	5.73/6.05	76,512/82,630	190572608
MB1806	Putative 30S ribosomal protein	Protein synthesis	Protein synthesis	3.0			4.44/4.79	66,234/72,144	1702195087
MB6403	Putative aminotransferase	Protein metabolism; Gluconeogenesis	Transamination	4.7	7	31	6.07/6.91	42,450/43,319	190575152
MB6405	Putative NADP-dependent alcohol dehydrogenase	Response to reactive oxygen species	Metal ion binding; Butanol dehydrogenase activity	4.3	4	17	5.71/6.93	38,658/41,493	190575829
MB7303	Putative threonine 3-dehydrogenase	Threonine catabolism	L-threonine 3-dehydrogenase activity	3.5	20	47	6.21/7.26	37,502/35,401	190572994
MB7502	Putative serine hydroxymethyltransferase	Glycine biosynthesis from serine	Glycine hydroxymethyltransferase activity	10.1	9	27	6.12/7.44	45,122/49,476	190572766
MB8406	Conserved hypothetical protein	Not known	Not known	7.8	22	59	7.15/5.99	43,882/85,923	190576469
MB8602	Putative biotin carboxylase	Fatty acid biosynthesis	ATP binding; Metal ion binding	5.2	13	34	6.55/7.97	49,609/60,821	190576068

^a^ Ratio of mean percentage integrated optical density for each protein from biofilm-grown and planktonic *S. maltophilia* cells (n = 6). ^b^ Number of tryptic peptides matched to protein sequence based on MS/MS spectra. ^c^ The percentage of amino acid coverage (peptides observed/theoretical number from sequence data given in *S. maltophilia* database). ^d^ Theoretical and practical pI of the protein. ^e^ Theoretical and practical mass, expressed in Da, of the protein. ^f^ GI accession (GenInfo identifier; sequence identification).

**Table 3 microorganisms-11-00442-t003:** Proteins (n = 8; signal > 1000 ppm; ≥ 3-fold *vs*. biofilm growth; *p* < 0.01, *t*-test) significantly downregulated in biofilm-grown *S. maltophilia* Sm126 cells.

Spot No.	Protein	Biological Process	Molecular Function	Fold Up-Regulation ^a^	Peptides Matched ^b^	Sequence Coverage ^c^	pI ^d^	MW ^e^	GI ^f^
MP0304	OmpA/MotB domain-containing protein	Ion transport	Porin activity	19.0	11	43	4.80/4.36	39,148/37,464	194364578
MP4106	Hypothetical protein Smal_0271	Not known	Not known	13.5	9	44	5.80/5.97	21,071/16,406	194364049
MP8009	Thioester dehydrase family protein	Fatty acid biosynthetic process	Dehydrase activity	3.6	8	43	4.90/8.21	17,497/7237	223039807
MP0011	Ribosomal protein L7/L12	Cytoplasmatic translation	Structural constituent of ribosome	4.0	4	50	4.79/4.51	12,589/10,677	254524344
MP0201	Thioredoxin	Removal of superoxide radicals	Detoxification	4.4	3	14	4.57/4.34	30,871/31,825	194366686
MP1006	50S ribosomal protein L9	Cytoplasmatic translation	Large ribosomal subunit rRNA binding	10.3	6	36	5.24/4.9	15,719/8238	190575039
MP8002	Putative sigma-54 modulation protein	Translation regulation	Translation regulation	5.5	2	26	6.64/8.17	11,739/10,127	190573137
MP8004	Cold shock protein	Response to stress	Nucleic acid binding	9.8	3	40	8.15/8.80	7573/7464	190573988

^a^ Ratio of mean percentage integrated optical density for each protein from planktonic and biofilm-grown *S. maltophilia* cells (n = 6). ^b^ Number of tryptic peptides matched to protein sequence based on MS/MS spectra. ^c^ The percentage of amino acid coverage (peptides observed/theoretical number from sequence data given in *S. maltophilia* database). ^d^ Theoretical and practical pI of the protein. ^e^ Theoretical and practical mass, expressed in Da, of the protein. ^f^ GI accession (GenInfo identifier; sequence identification).

## Data Availability

Data is contained within the article or Appendix A.

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
