# Peer review of "Comparative Proteomic Analysis of Protein Patterns of Stenotrophomonas maltophilia in Biofilm and Planktonic Lifestyles"

_microorganisms, 2023, doi:10.3390/microorganisms11020442_

Round 1

Reviewer 1 Report

The manuscript Proteomic analysis of Stenotrophomonas maltophilia biofilm  formation reveals proteins unique to the biofilm lifestyle

The authors reported the first proteomic analysis of an S. maltophilia biofilm, providing initial insight into its cellular biology I have no comments regarding the manuscript itself, or its organization. In my opinion, the article in the presented version is suitable for publication after minor revisions. General comments -  Is it possible to talk more comprehensively about S. maltophilia is an environmental bacterium ( introduction section).

-     authors have been used the technique of Maldi TOF MS to identify proteins , if any identified proteins through that technique .

Author Response

Please see the attached file containing point-by-point replies.

Besta regards

Reviewer 2 Report

The title of Manuscript should be reframed

Line 95: What is PBS

Line 88-90: reframe sentence

Line 98: Use Petri dish or plate

Line 98-103: Rewrite in simple, confusing lines

Line 106: Benzonase 0,25U/µl; correct 0.25U/µl

Line 107: Incubation time at room temperature

Line 109: Check whether 50,000xg or 5,000x g for 30 min)

Section 2.1 : Heading and section should be separate and not in italics

Uniform same font in overall MS

All the figure having very low resolution, improve all figures resolution

Whole MS is in plagiarism i.e Abstract, Material method, Result discussion and conclusion

Reduce the % of plagiarism upto 10%

All genus and species name should be be in Italics

The writing could be improved by strengthening the connectivity between paragraphs. There are several places where new topics are introduced and connections to the previous subject are not clear. Read whole manuscript and correct wherever required.

Introduction:

The introduction does not clearly state the purpose of the research – please amend.

Conclusions

The conclusions are too general, format according to future aspects. Please make them more specific.

Carefully read whole manuscript line by line and improve the sentence formation

Cross check all references and style of reference according to Journal format, use abbreviation of journal name in reference

Author Response

Please see the attached file containing point-by-point replies.

Best regards

Reviewer 3 Report

In this work, the authors analysed the proteome of the bacteria Stenotrophomonas maltophilia, in particular a clinical isolate from a patient with Cystic Fibrosis able to produce biofilms. To evaluate the proteins involved in biofilm formation the authors analysed the protein profile of planktonic and biofilm S. maltophiliacells, and compared the differentially expressed proteins through 2D electrophoresis, followed by protein identification by MALDI-TOF-MS and N-terminal sequencing. 

Overall the manuscript is well-written, the results are interesting, revealing a set of proteins only expressed in biofilms, or upregulated in this situation, pinpointing some potential targets for drug development. 

Introduction

Line 70: please indicate the reference of the studies that have compared the protein expression profiles between biofilm and planktonic S. maltophiliacells mentioned here.

Materials and Methods

Line 88-89: This sentence is not clear: “The culture was diluted and added to 560ml of TSB.” What is the final volume, and consequently the final OD?

Line 99: What does TC stand for in TC-treated dishes?

Line101. Is the pH of the PBS also 7.3?

Line 107: Please state what RT stands for.

Line 114: Please state what 2-DE stands for.

Line 123: Please state what IPG stands for.

Line 149 The information on the database used should be included (reference or website)

Line 165: The samples were not submitted to coating before observation in the SEM?

Results

Line 176. The crystal violet colorimetric assay should be included in the material and method section, and not in the result section.

Figure 1. The different panels should be labelled A) and B) and the figure legend should include that nomenclature. Also, in the panel for planktonic growth, the absorbance values presented in the y-axis should be presented in a logarithmic scale, to allow the visualisation of the linearity present in the exponential growth phase. If the results are the mean OD values, the standard deviation values should be presented in the figure.

Figure 2. The images should be improved by removing unnecessary information and presenting just the scale on the inferior right section. For clarity, the scale could have a dark background, or a bigger size letter to facilitate the reading. 

Line 217: In situations where no information was found in the genomic database of S.maltophilia, could you find other similar proteins from other organisms? This is what is stated in line 228.

Line 259: I am not sure that the detection of a single enzyme, is sufficient to state the presence of a quorum sensing system. Maybe the sentence could say “Suggesting the presence of a QQ System”.

Line 315: This sentence is not clear.

Line 335: Italic is missing in the species name.

Line 500: Please correct the title. 

The discussion is very extensive, and sometimes it is difficult to determine which are the most relevant aspects found in this work. Maybe the authors could include a final figure where they summarize the major actors and pathways for biofilm formation in S. maltophilia found this work.

Author Response

(The authors gave the same response as above.)

Round 2

Reviewer 2 Report

Accepted in present form